# Reproducibility Study of "Vision Transformers Need Registers"

## Abstract

Vision Transformers (ViTs) have achieved State-Of-The-Art (SOTA) performance in numerous tasks. However, the emergence of high-norm artifact tokens in supervised and self-supervised ViTs hinders interpretability of attention maps of such models. This study reproduces and validates previous work (5) addressing this issue through the use of register tokens - learnable placeholders added to the input sequence - that mitigate artifacts and yield smoother feature maps. We evaluated the presence of artifacts in various ViT models, namely DeiT-III and DINOv2 architectures, and investigated the impact of fine-tuning pre-trained ViTs with register tokens and additional regularization introduced. By conducting experiments on pre-trained and fine-tuned models, we confirm that register tokens eliminate artifact and improve attention map interpretability.

## 1 Introduction

Vision transformers (6) are commonly trained on label (14) or text (12) supervision, and have emerged as a robust framework for learning visual representations, although despite their success, recent studies highlight the presence of artifacts in feature maps of supervised and self-supervised ViTs (5). These artifacts, characterized by high-norm tokens, can affect dense prediction tasks and compromise the interpretability of attention maps. To address this issue, Darcet et al. (5) proposes to augment the input sequence with additional learnable tokens, termed *register tokens*. These tokens serve as placeholders for computations, effectively eliminating artifacts and enabling smoother feature maps. This innovation, first introduced in NLP (1), improves performance in dense prediction tasks, as well as enhances compatibility with unsupervised object detection methods.

In this study, our objective is to reproduce and validate these findings. Specifically, we investigate the presence of artifacts in various ViT models (DeiT-III (14), DINOv2 (10)) and evaluate the efficacy of the register token approach. Additionally, inspired by work on transformer fine-tuning for NLP (7; 9), we explore fine-tuning frameworks with added register tokens, with and without L2-norm regularization, as a more efficient alternative to full pre-training for reducing artifacts. Fine-tuning, as opposed to retraining the entire model end-to-end, is significantly less resource intensive, reducing computational costs, memory load, and training time, and therefore being more environmentally friendly by lowering the overall energy consumption (11).

## 2 Scope of reproducibility

The work addresses a problem in ViTs, specifically the presence of high-norm artifact tokens. These artifacts, which emerge during training in low-informative background regions, are repurposed for internal computations, leading to less interpretable attention maps. Darcet et al. (5) introduce a simple yet effective method that adds register tokens to the image token sequence, independent of the input image. This approach successfully mitigates artifacts and enhances performance across various tasks. Originally proposed in the NLP domain to improve translation tasks (1), register tokens also enhance the quality and interpretability of attention maps, enabling transformers to achieve their expected performance in downstream tasks such as

detection (3), segmentation (16), and monocular depth estimation (10). Building on the authors' proposed incorporation of registers into the model, a fixed number N (originally set to 4) of new tokens were added to the sequence. These learnable tokens were introduced after the patch embedding layer, similar to the `[CLS]` token, and were initialized from a normal distribution. At the end of the ViT, the register tokens were discarded, leaving only the image patch tokens and the `[CLS]` token for image representation. Figure 1 illustrates how the registers are integrated into the model.

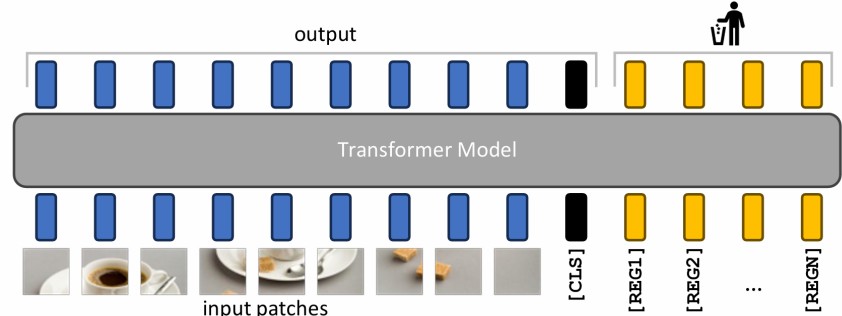

Figure 1: Illustration of the solution (5): N additional learnable input tokens (shown in yellow) are introduced, which the model utilizes as registers. At the model's output, only the patch tokens and the `[CLS]` token are retained for both training and inference.

The main claims tested in this work are:

- **Claim 1**: Artifact tokens are high-norm outliers.

- **Claim 2**: Artifact tokens emerge in sufficiently large ViT models after extended training.

- **Claim 3**: Artifact tokens encode global image information with minimal local detail.

- **Claim 4**: High-norm tokens appear in regions with redundant patch information, on patches similar to their neighbors.

- **Claim 5**: Register tokens remove high-norm artifacts, smoothing feature maps, improving interpretability and dense prediction tasks performance.

## 3 Methodology

To reproduce the results and evaluate the claims made in the paper (5), we relied on public implementations and pre-trained models, as the original code incorporating register tokens was not made open source. Specifically, we used a public implementation[1], which we adapted for our research questions, further elaborated on later in this work. For both DeiT-III (14) and DINOv2 (10) models, we reproduced most of the experiments described in the paper. For DINOv2, pre-trained configurations with and without register tokens were publicly available, enabling us to directly compare their performance. Additionally, since there was no pre-implemented function to access attention maps, we implemented it ourselves, taking inspiration from another repository[2]. Since models trained with register tokens require training from scratch, we explored whether artifacts could be mitigated in a pre-trained model. To this end, a pre-trained DeiT-III-small model was fine-tuned for image classification with an added L2-norm loss to penalize high-norm artifacts in the final attention layer. The official DeiT codebase[3] was modified to incorporate register tokens, applying regularization to suppress artifact tokens. This approach aimed to assess whether fine-tuning with regularization could effectively reduce artifacts while preserving model performance, providing a more computationally

---

[1] `https://github.com/kyegomez/Vit-RGTS`

[2] `https://gitlab.com/ziegleto-machine-learning/dino`

[3] `https://github.com/facebookresearch/deit`

efficient alternative by leveraging pre-trained models instead of full retraining. The codebase of our work is available under the following link: `https://github.com/vit-reg/vit-reg`.

### 3.1 Model descriptions

The original paper (5) focused on three distinct training paradigms while examining the presence and role of high-norm tokens: supervised training for classification, self-supervised methods for learning visual features, and text-image aligned models. Of these three paradigms, we examined the first two. Detailed descriptions of the models used in our experimental setup are provided below.

**DeiT-III** (14) is a supervised ViT model optimized for image classification. It is designed to be efficient and scalable while maintaining high performance on large datasets. Unlike its predecessors, DeiT-III simplifies the training process by adopting a streamlined data augmentation strategy, utilizing only three augmentations, which aligns more closely with practices in self-supervised learning. This approach not only reduces complexity but also improves training efficiency. Additionally, it exhibits strong performance in transfer learning and semantic segmentation tasks, which makes it a versatile model for various computer vision applications.

**DINOv2** (10) is a self-supervised ViT model that builds upon its predecessor, DINO (4), to provide advanced visual features for a wide range of tasks. While DINO focuses on learning meaningful features directly from image data by identifying important regions, DINOv2 enhances this framework to excel in more complex tasks such as semantic segmentation and depth estimation. DINOv2 leverages improved architectural designs and larger-scale training, making it a powerful tool for high-level vision applications without requiring labeled data.

| Model | Type | Parameters |
|---|---|---|
| DeiT-III | Small/16 | 22M |
| DeiT-III | Base/16 | 86M |
| DINO | Big/16 | 85M |
| DINOv2 | Large/14 | 300M |
| DINOv2 | Giant/14 | 1,100M |

Table 1: Types of models and their parameters.

We reproduced and evaluated the following ViTs, all pre-trained on ImageNet-21k (13) and initialized with publicly available pre-trained weights (Table 1). Note: The number of parameters added when using register tokens is negligible, as the exact amount is given by $N \times P$, where $N$ represents the number of tokens and $P$ represents the patch dimension.

### 3.2 Datasets

- **Cifar-10 (8)**: A benchmark dataset containing 60,000 images, each with a resolution of $32 \times 32$ pixels and evenly distributed across 10 classes. For the linear and logistic regression experiments with DINOv2, we selected 800 images for training and 200 for testing, all resized to $196 \times 196$ pixels. For logistic regression experiments with DeiT-III, we used 1,500 training images and 400 testing images, resized to $384 \times 384$ pixels. For linear regression experiments with DeiT-III, we selected 1,000 training images and 10 testing images, also resized to $384 \times 384$ pixels.

- **ImageNet (13)**: A widely used large-scale dataset for image classification, containing over 1.2 million images across 1,000 classes in the ImageNet-1k version and 14 million images across more than 21,000 classes in the ImageNet-22k version. For our experiments with DINO, DINOv2, and DeiT-III models, we used images from the ImageNet-22k dataset, resized to $384 \times 384$ pixels. For fine-tuning, we used the ImageNet-1k dataset.

- **ADE20K (17)**: A dataset of over 25,000 images annotated with pixel-wise semantic segmentation labels across 150 categories, including objects and background regions. For our experiments with DINOv2, and DeiT-III models, we used the official training and validation splits. All images were resized to 518x518 pixels for DINOv2 and 224x224 pixels for DeiT-III.

## 3.3 Experimental setup

We reproduced key experiments from the original paper on DeiT-III (14), DINO (4) and DINOv2 (10) models. For each model without registers added, we plotted the attention maps of the last transformer layer and the feature norms of patch tokens to investigate the presence of artifacts, as observed in DINOv2 and DeiT-III models but not in DINO. For DINOv2 and DeiT-III, we focused on identifying artifacts in low-informative regions, examining whether high-norm tokens corresponded to high-attention values. To analyze the redundancy of patch information, we identified high-norm tokens and computed their cosine similarity with neighboring tokens. To assess whether these tokens aggregate global information, we trained a logistic regression classifier on patch token embeddings, using classification accuracy to evaluate the representational capacity of high-norm versus normal tokens. Additionally, we visualized attention maps across layers to track how global information evolved throughout the network. For DINOv2 models with register tokens, we repeated these analyses to verify whether registers eliminated artifacts, as claimed in the paper. These experiments combined qualitative analyses (visualizations of attention maps and feature norms) with quantitative metrics (cosine similarity, classification and segmentation performance) to provide a comprehensive evaluation of the models' behaviors.

Considering the fine-tuning pipeline, the original DeiT-III codebase was modified to include the addition of register tokens, with the option to load pre-trained DeiT-III model weights from the original setting. Furthermore, regularization of the L2-norm was introduced to penalize high values from the last attention block. Specifically, the mean L2-norm of image patches from the last attention block was added to the cross-entropy loss. The L2-norm loss was scaled by an adjustment parameter, $\lambda$. The overall loss function can be expressed as:

$$\text{Loss} = \text{CrossEntropyLoss} + \lambda \cdot \text{L2NormLoss}. \tag{1}$$

To prevent the feature map values of all tokens from approaching zero, alternative $\lambda$-schedulers were explored. The penalization term was gradually decreased over time using four different decay functions: harmonic decay, exponential decay (both hard and soft versions), and step decay.

- **Harmonic decay**: $\lambda(epoch) = \lambda/(epoch + 1)$

- **Exponential decay (hard)**: $\lambda(epoch) = \lambda^{[(epoch//10)+1]}$

- **Exponential decay (soft)**: $\lambda(epoch) = \lambda^{[(epoch/10)+1]}$

- **Step decay**: $\lambda(epoch) = \lambda$ if $epoch < 10$ else $0$

Additionally, to further investigate the properties of artifact tokens, we removed them at a specific layer during inference to assess whether they reemerge or if their absence results in smoother attention maps, providing insights into their role in ViTs.

## 3.4 Hyperparameters

The hyperparameter values were determined based on task-specific requirements and computational constraints.
For DINOv2 models, logistic regression experiments for artifact and normal tokens classification used 1000 iterations, a high-norm threshold of 100, and seeds 42, 123, and 999. Position prediction via linear regression used the same seeds and high-norm threshold, with a batch size of 10. Performance regression was also run with a 1000-iterations limit. The linear head for segmentation was trained for 20 epochs, using batch size of 32. For the DeiT-III small model, logistic regression experiments for artifact and normal token classification

used a batch size of 16, 1000 iterations, and a seed of 42, while the position prediction experiment utilized followed the same settings.

Fine-tuning hyperparameters for DeiT-III Small (12 blocks) included varying the number of frozen blocks (8–11), the $\lambda$ parameter for L2-norm regularization (0, 0.1, 0.01), and the number of register tokens (0, 1, 4). Configurations without registers and L2-norm regularization were omitted as they match standard DeiT-III training. In total, 32 fine-tuning experiments were conducted. Due to computational constraints, each configuration was trained for 10 epochs on ImageNet-1k, ensuring a balanced yet diverse exploration of hyperparameter settings. Considering segmentation, a linear head along with the last attention block were trained for 50 epochs using batch size of size 32.

## 3.5 Computational Requirements

The DINO and DINOv2 experiments were run on a CPU for approximately 15 hours and primarily involved inference, with no model training. Similarly, the DeiT-III reproducibility experiments were conducted locally using Apple's MPS backend and lasted around 1 hour in total. The only training performed during these experiments was the training of lightweight logistic regression classifiers on extracted image features, which was computationally negligible and executed on a CPU.

In contrast, the segmentation experiments for DINOv2 and DeiT-III, as well as the fine-tuning experiments for DeiT-III, required significantly more computational resources. The experiments were run on an NVIDIA A100 and H100 SXM4 80 GB GPU with a TDP of 400W, using private infrastructure provided by SURF.nl. The classification fine-tuning experiments with DeiT-III had a cumulative runtime of under 65 hours.

For the segmentation task, linear probing of DINOv2 involved 6 experiments, each lasting approximately 3 hours, totaling 18 hours. For DeiT-III, 12 segmentation fine-tuning experiments were conducted, each lasting approximately 28 minutes, resulting in a total of less than 6 hours. Table 2 summarizes the computational cost for these fine-tuning experiments.

| Experiment | Type | Hardware | Runtime (h) |
|---|---|---|---|
| DeiT-III (classification fine-tune) | Small/16 | A100 | 64h |
| DeiT-III (segmentation head training) | Small/16 | A100 | 6h |
| DINOv2 (segmentation head training) | Large/14 | H100 | 18h |

Table 2: Computational requirements and estimated emissions for DeiT-III fine-tuning experiments. The average time per classification experiment was slightly above 2 hours, resulting in a total runtime of under 65 hours.

## 4 Results

For the DINO (4) and DINOv2 (10) models, we successfully reproduced most of the experiments in the original paper (5), with the results supporting all the claims made. The same applies to the DeiT-III (14) model, except that artifacts were observed even in the small version of the model. The following sections provide a detailed overview of all the experiments conducted, along with a discussion on fine-tuning the pre-trained DeiT model with registers. For the DINO and DINOv2 models, we used a resolution of 952x952 pixels, while for the DeiT-III model, we used 384x384 pixels.

### 4.1 Results reproducing original paper

### 4.1.1 Attention maps visualization

For DINOv2 Large, we observed a single pixel near the boundary as an outlier on the combined attention map (averaged across heads), potentially an artifact (Figure 2c). For DINOv2 Giant, similar results were obtained, with boundary artifacts visible in the attention maps (Figure 2d). In addition, we visualized the

attention maps in the logarithmic scale to better depict which parts of the image were attended (Figure 15). These findings align with the observations in the original paper. For models with register tokens, the artifacts were no longer present (Figures 2h 2i) and ther attention maps showed a clean distribution, with no high-attention outliers near the boundaries. This confirms claim 5. Unlike DINOv2, the attention maps of the DINO model do not exhibit artifacts (Figure 2b). These findings align with the paper's claim that DINO produces clean attention maps.

Considering DeiT-III, high-norm tokens were found in both the small and base versions of the model (Figures 2e, 2f). Upon examination of the number of parameters in Table 1, it can be seen that even models with a smaller number of parameters may exhibit the presence of artifacts. This finding contradicts claim 2, as shown in Figure 2e, and suggest that the occurrence of artifacts is not solely dependent on the size of the model.

**Without Registers**

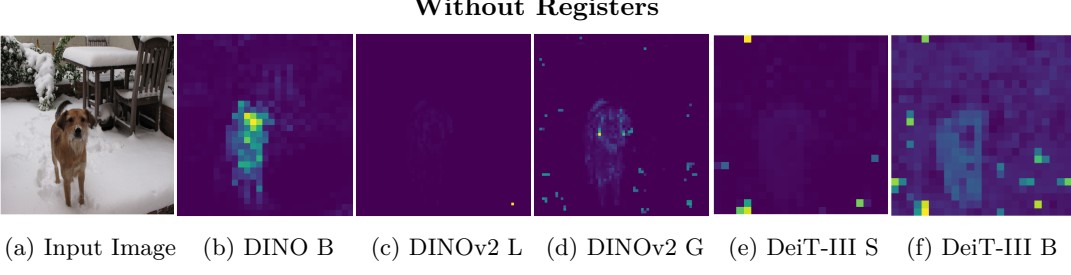

(a) Input Image   (b) DINO B   (c) DINOv2 L   (d) DINOv2 G   (e) DeiT-III S   (f) DeiT-III B

**With Registers**

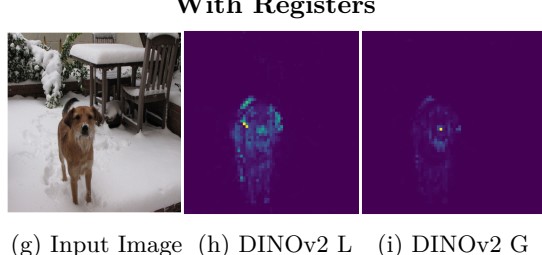

(g) Input Image   (h) DINOv2 L   (i) DINOv2 G

Figure 2: Attention maps for DINO, DeiT-III Small and Base, and DINOv2 Large and Giant models, with and without register tokens. DINO Big model does not exhibit artifacts without register tokens, whereas artifacts are present in all DINOv2 and DeiT-III models. With the inclusion of register tokens, artifacts disappear in the DINOv2 models. Note that attention maps for DeiT-III models with register tokens are not shown, as pretrained versions of these models are not publicly available. Fine-tuned versions of DeiT-III with register tokens are included in Figure 13

.

### 4.1.2   Norm values distribution

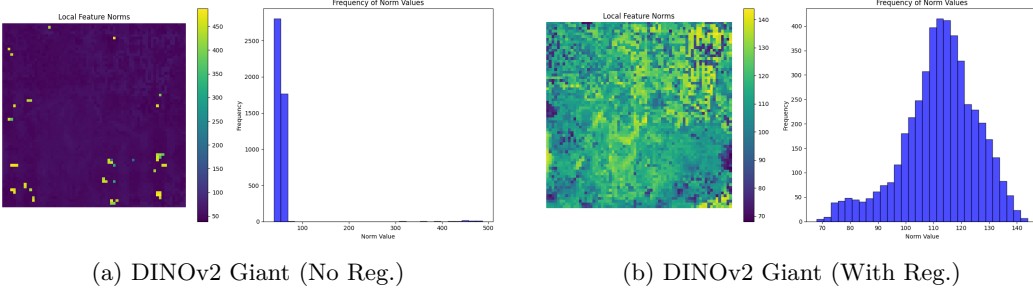

(a) DINOv2 Giant (No Reg.)                    (b) DINOv2 Giant (With Reg.)

Figure 3: Feature norm distributions for the DINOv2 Giant model, with and without register tokens.

The feature norms plots for DINOv2 models showed high-norm outliers. As expected, DINO and DINOv2 with registers feature norms analysis revealed less high-norm outliers (Figure 16). This can be specifically observed in Figure 3. DeiT-III feature norms (Figure 17) indicate a small fraction of higher feature norms, with elevated values often observed near the edges of the image. The presence of these higher-norm tokens does not appear to result in a bimodal distribution, as their occurrence is limited. The plot of norms across the layers of DeiT-III Small reveals the presence of high norm values starting from layer 4 (Figure 4). These findings further confirm the existence of artifacts within this small model, directly contradicting claim 2.

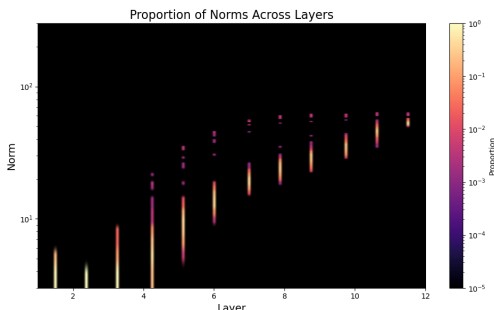

Figure 4: Visualization of the distribution of feature norms across layers for DeiT-III Small, confirming the presence of artifacts.

### 4.1.3 High-norm high-attention correspondence

For DINOv2 and DEIT-III models, the top 5% tokens with highest attention values (represented as red dots in Figures 5b, 5c and 5d) were confirmed to correspond to high-norm artifact tokens, validating claim 1.

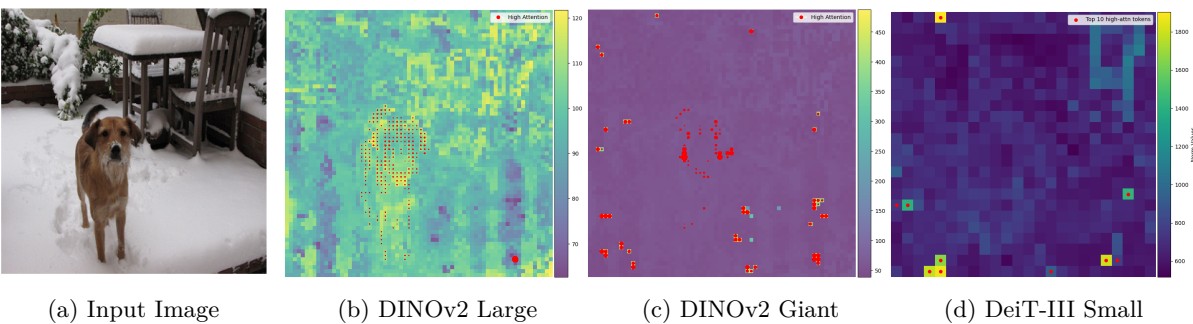

(a) Input Image      (b) DINOv2 Large      (c) DINOv2 Giant      (d) DeiT-III Small

Figure 5: High-attention and high-norm tokens correspond in all models. In the first two plots, the size of the red dots represents the magnitude of attention for each token, highlighting regions with the strongest focus (see Appendix for more example images).

### 4.1.4 Cosine similarity between high-norm tokens and their neighbors

The analysis of the distribution of cosine similarity between input patches and their four neighboring patches, with separate plots for artifact patches and normal patches is shown in Figure 6, highlighting the differences in their behavior. Cosine similarity analysis on DINOv2 Large revealed that high-norm tokens, defined as the top 5% output tokens with high norms, predominantly appeared in regions with redundant patch information, supporting claim 4 (Figure 6a). Similarly, analysis on DeiT-III Small, also supports claim 4. (Figure 6b).

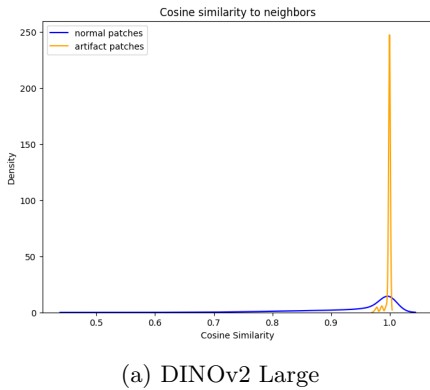
(a) DINOv2 Large

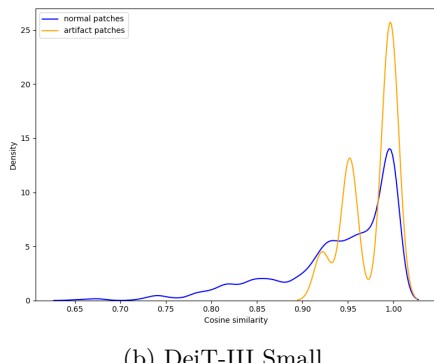
(b) DeiT-III Small

Figure 6: Cosine similarity for DINOv2 Large and DeiT-III Small models. High-norm tokens, exhibit higher similarity to their neighbors compared to normal patches.

### 4.1.5 Artifacts global information aggregation

To assess the global information stored in high-norm tokens, logistic regression was applied for both DINOv2 and DeiT-III models without register tokens. In both cases, high-norm tokens consistently achieved higher classification accuracy than normal tokens (Table 3). These findings support claim 3, demonstrating that high-norm tokens are artifacts and contain more global information.

| Tokens | DINOv2 L | DINOv2 G | DeiT-III S |
|---|---|---|---|
| High-Norm Tokens | 95.63% | 97.71% | 84.67% |
| Normal Tokens | 87.92% | 84.38% | 78.67% |

Table 3: Logistic Regression Results for High-Norm and Normal Tokens on CIFAR-10.

For DINOv2 Large, the attention maps show that intermediate layers distribute attention across the entire image. However, in the last few layers, a single pixel with high attention emerges, corresponding to an artifact. Specifically, these artifacts appear at layer 18 of 23 and hold global information, as verified through additional analysis. For DINOv2 Giant, artifacts emerge at layers 15–16 out of 40, as specified in the original paper. These correspond to tokens that hold global information, as their attention spans the entire image in previous layers (Figure 18).

For DeiT-III Small, we also visualized the attention maps across layers, observing that artifacts emerge around block 5 out of 12. Additionally, we plotted the attention maps for the [CLS] token, a randomly selected patch token with low norm, and a high-norm artifact token. The attention maps for the [CLS] and artifact tokens reveal that, despite the presence of artifacts, some attention is focused on the main subject of the image. In contrast, the attention map for the patch token lacks focus on the subject, suggesting that artifact tokens capture global information, whereas patch tokens do not (Figures 19).

### 4.1.6 High-norm tokens hold little local information

To evaluate whether high-norm tokens retain local information, we trained a linear classifier to predict the original spatial position of each patch token. We extracted patch token embeddings and their corresponding 2D positions. Tokens were then grouped by their L2-norm into high-norm and normal-norm subsets. A separate linear classifier was trained for each group to predict the position of each patch token in the image, and we reported top-1 accuracy and average positional distance over three random seeds. The results for DINOv2 Large (Table 4) without registers show that high-norm tokens have lower top-1 accuracy compared to normal tokens, suggesting that they hold little positional information. The results align with the observations from the original paper and further support Claim 3.

For the DeiT model, we confirmed that high-norm patches are artifacts, as they capture image category information but lack positional information. As shown in Figure 7, normal tokens achieve higher position prediction accuracy than high-norm tokens, further supporting Claim 3.

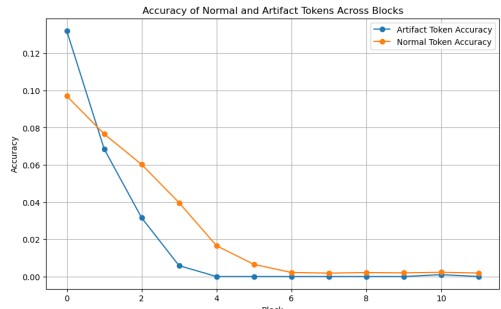

Figure 7: Position prediction accuracy across layers for DeiT-III Small on CIFAR-10. Artifact tokens consistently show lower accuracy than normal tokens.

| Tokens | Top-1 Accuracy | Average Distance |
|---|---|---|
| High-Norm Tokens | 1.53% | 5.09 |
| Normal Tokens | 13.82% | 1.38 |

Table 4: Position prediction results for DINOv2 Large on CIFAR-10

### 4.1.7 Impact of the register tokens on image representation quality

To assess whether the use of register tokens affects the representation quality of features, we trained a logistic regression classifier on features extracted from CIFAR-10 images to evaluate the discriminative power of the representations learned by each DINOv2 model. The results indicate that the use of register tokens does not degrade representation quality. On the contrary, a slight increase in classification accuracy is observed when registers are included in the models (Table 5). These findings confirm that register tokens preserve or improve representation quality.

| Registers | DINOv2 Large | DINOv2 Giant |
|---|---|---|
| Without Registers | 97.50% | 97.00% |
| With Registers | 99.50% | 97.50% |

Table 5: Performance Regression Results for DINOv2 Large and Giant on CIFAR-10.

### 4.1.8 Segmentation analysis

We evaluate the segmentation performance of DINOv2-L and DeiT-III-Small models, both with and without register tokens, on the ADE20K dataset. For DINOv2, we use the official ViT-L/14 backbones (with and without registers), keeping them fully frozen during training and training only a linear segmentation head

on top (linear probing). For DeiT-III-Small, we use the official backbone and similarly train a linear head in the no-registers setting. When introducing register tokens into DeiT-III, we unfreeze the final attention block and add register tokens at that level. For both models, we compute the mean Intersection over Union (mIoU) per class to quantify segmentation quality. The segmentation head is trained for 20 epochs for DINOv2 and for 50 epochs for DeiT-III.

The results, averaged over three runs, are summarized in the tables below 7 6:

| Registers | mIoU (%) | Variance |
|:---:|:---:|:---:|
| 0 | 23.72 | $2.39 \times 10^{-4}$ |
| 1 | 23.95 | $2.12 \times 10^{-4}$ |
| 2 | **24.06** | $3.76 \times 10^{-5}$ |
| 4 | 24.01 | $1.23 \times 10^{-4}$ |

Table 6: Segmentation results for DeiT-III-Small on ADE20K with varying numbers of register tokens.

| Registers | mIoU (%) | Variance |
|:---:|:---:|:---:|
| 0 | 41.49 | $4.34 \times 10^{-8}$ |
| 4 | **43.37** | $6.34 \times 10^{-8}$ |

Table 7: Segmentation results for DINOv2-L on ADE20K with and without register tokens.

It is clear from the results that the introduction of register tokens does not degrade segmentation performance for both architectures. These are strong results given our setup: we train only lightweight segmentation heads and, in the case of DeiT-III, do not use a pretrained backbone with register tokens. Despite this, we still observe consistent improvements, especially for DeiT-III, where even the partial addition of register tokens improves performance. This suggests that register tokens can be beneficial even without end-to-end pretraining.

## 4.2 Results beyond original paper

Our findings indicate that contrary to the claim made in the original paper that artifacts only appear in sufficiently large ViTs, we show that they also emerge in models of smaller size. This can be seen from the attention maps of a pre-trained DeiT-III Small in Figure 2e.
Furthermore, we also show that incorporating register tokens and L2-norm regularization during fine-tuning reduces artifacts in attention maps, resulting in smoother and more focused attention. Despite most layers remaining frozen, attention values began to spread within the trainable blocks, indicating that this could be a more efficient alternative to training a model from scratch.

### 4.2.1 Fine-tuning

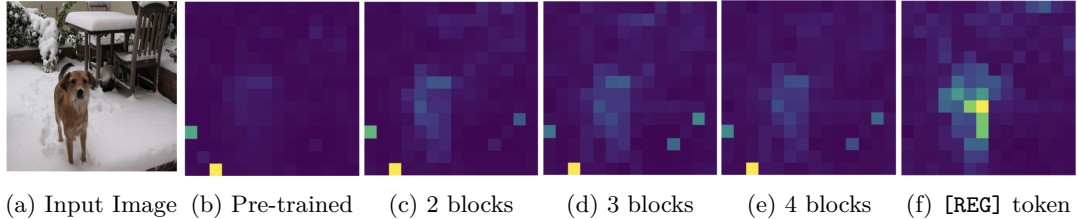

(a) Input Image  (b) Pre-trained  (c) 2 blocks  (d) 3 blocks  (e) 4 blocks  (f) `[REG]` token

Figure 8: Attention maps for the `[CLS]` token in the last attention block for a pre-trained DeiT-III model and fine-tuned versions with register tokens, varying the number of unfrozen attention blocks. The final image shows the attention distribution of a register token with 3 unfrozen attention blocks.

The quantitative results of the fine-tuning experiments are presented in Table 8. Following the original paper, we use 4 registers, as this number provides a good balance between performance and computational cost across various tasks. Additionally, we experimented with 1 registers to explore how performance is affected by changing the number of registers.

The addition of registers does not harm model performance compared to fine-tuning without them, suggesting they are a viable extension that preserves the model's core behavior. However, since the performance differences with 1 register were minimal, only results for 0 and 4 registers are reported in the table. Overall, no significant performance variation was observed due to the number of registers.

The table also shows that both top-1 and top-5 accuracy tend to decrease as more layers are unfrozen and the fixed $\lambda$ regularization parameter increases. While a performance drop due to L2-norm regularization is expected, it is noteworthy that this decline is relatively small across all cases. This suggests that, although regularization slightly reduces accuracy, the overall degradation remains limited - even after fine-tuning.

Additionally, the effect of fine-tuning the entire model was examined, considering three selected settings (one register introduced with $\lambda \in \{0, 0.01, 0.1\}$) and training for 10 epochs. The results show that the top-1 accuracy varies between 75% and 78%, while the top-5 accuracy ranges from 92% to 94%. Comparing these values with the settings where the first $n$ attention blocks were frozen ($n \in \{8, 9, 10, 11\}$), we observe a drop in performance, and the artifacts, although the attention maps began to spread more within the image, have not been fully removed. We show the effect of fine-tuning on artifacts.

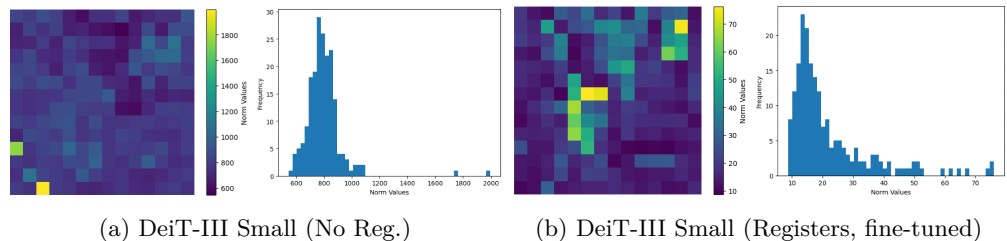

(a) DeiT-III Small (No Reg.)          (b) DeiT-III Small (Registers, fine-tuned)

Figure 9: Feature norm distributions for DeiT-III Small, comparing the pretrained model (a) and a fine-tuned model with registers (b). In the pretrained case, high-norm tokens are outliers and appear in uninformative background areas. After fine-tuning, they become spatially focused on meaningful objects (e.g., dog, chair, table), and the overall distribution is continuous and regularized.

For the qualitative analysis, we examine the progression of attention values across blocks, focusing on the [CLS] and the register tokens. Figure 8 presents the attention distribution in the last block and Figure 13 illustrates how attention values evolve in the pre-trained DeiT-III small model compared to its fine-tuned versions without L2-norm regularization (configurations ranging from 2 to 4 unfrozen layers). Notably, as more layers are unfrozen, the attention values become more spread across the image, in contrast to just the pre-trained model, where they are predominantly concentrated on a single point. This trend suggests that increasing the number of unfrozen layers facilitates broader attention spreading within the layers. Additionally, the highest attention values are often located near the center of the image, likely influenced by the centered composition of images in the ImageNet dataset. When registers are introduced with L2-norm regularization, the last attention block tends to focus primarily on the object without introducing artifacts - the spread of attention values is more evenly distributed, with a shift toward the center of the image, which is desirable. Figure 22 illustrates the effect on attention values in the last two trainable attention blocks.

| Layers Frozen | Registers | Parameters | L2-Norm $\lambda$ | Test Acc@1 (%) | Test Acc@5 (%) |
|---|---|---|---|---|---|
| **no fine-tuning** | | | | **82.81** | **96.72** |
| 11 | 0 | 2,236,648 | 0.01 | 82.26 | 96.27 |
| | | | 0.10 | 81.06 | 95.65 |
| | 4 | 2,238,184 | 0.00 | **82.42** | **96.38** |
| | | | 0.01 | 82.22 | 96.29 |
| | | | 0.10 | 80.44 | 95.20 |
| 10 | 0 | 4,011,880 | 0.01 | 81.88 | 96.07 |
| | | | 0.10 | 81.55 | 95.83 |
| | 4 | 4,013,416 | 0.00 | **82.28** | **96.40** |
| | | | 0.01 | 81.94 | 96.08 |
| | | | 0.10 | 81.62 | 95.84 |
| 9 | 0 | 5,787,112 | 0.01 | 81.33 | 95.84 |
| | | | 0.10 | 80.71 | 95.52 |
| | 4 | 5,788,648 | 0.00 | **82.09** | **96.30** |
| | | | 0.01 | 81.34 | 95.88 |
| | | | 0.10 | 80.95 | 95.56 |
| 8 | 0 | 7,562,344 | 0.01 | 80.24 | 95.45 |
| | | | 0.10 | 79.64 | 95.09 |
| | 4 | 7,563,880 | 0.00 | **81.83** | **96.15** |
| | | | 0.01 | 80.51 | 95.57 |
| | | | 0.10 | 78.43 | 94.45 |

Table 8: Hyperparameter tuning results for fine-tuning the DeiT-III small model on ImageNet-1K. The first row represents the accuracy values for the pre-trained DeiT-III small model. The highest accuracy values within each number of frozen layers are shown in bold.

As illustrated in Figure 22, the qualitative results of fine-tuning with ten fixed layers appeared promising (higher attention values moved from the background to the center of an object); therefore, we increased the number of epochs to 50 to evaluate whether longer training would lead to improved performance and smoother attention maps. Additionally, to prevent the feature map values from approaching excessively low values that would result in the loss of its qualitative information, we explored alternative $\lambda$-schedules. As shown in Figure 10, the decay strategies outlined in Section 3.3 yield similar results between each other - the attention maps of the register tokens clearly capture the shape of the subject in the image, whereas the attention map of the `[CLS]` token became hard to interpret, i.e. the artifacts reappeared. This suggests that further research is needed to investigate alternative high-norm penalization schedules or explore different techniques.

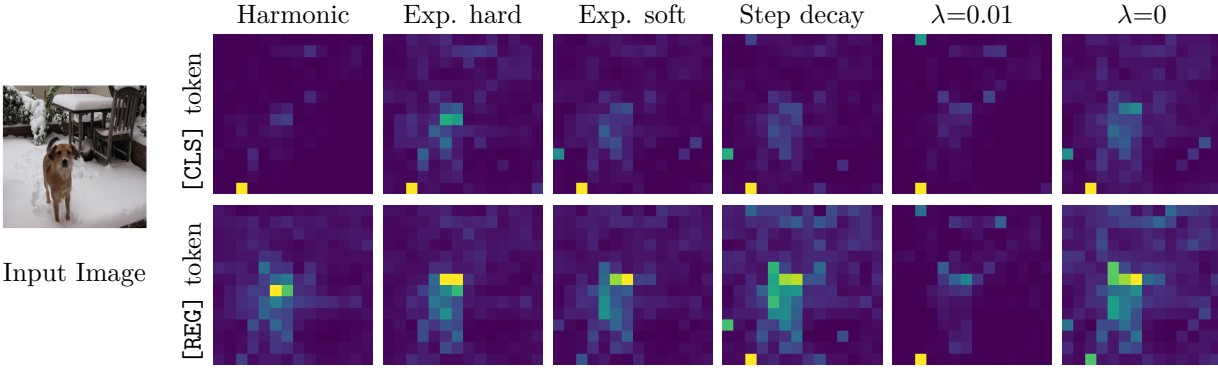

Figure 10: Comparison of `[CLS]` and the register (`[REG]`) tokens' attention values from the last attention block of a fine-tuned DeiT-III small model after 50 epochs using different lambda schedules: harmonic decay ($\lambda(0) = 0.1$), hard exponential decay ($\lambda(0) = 0.1$), soft exponential decay ($\lambda(0) = 0.01$), step decay ($\lambda(0) = 0.01$), $\lambda = 0.01$, and $\lambda = 0$ (see Appendix for more example images).

### 4.2.2 Additional experiment: manual removal of artifacts

To investigate the role and significance of artifact tokens in comparison to other tokens, we conduct an experiment where artifact tokens are manually removed at each layer during inference (Figure 11). The objective is twofold: (1) To determine whether the removed artifact tokens reemerge during the model's forward pass or if their removal leads to smoother attention maps. (2) To gain deeper insights into the contribution of artifact tokens during inference of Visual Transformers (ViTs).

Further analysis of the experiments can be found in Section 6.2 of the Appendix.

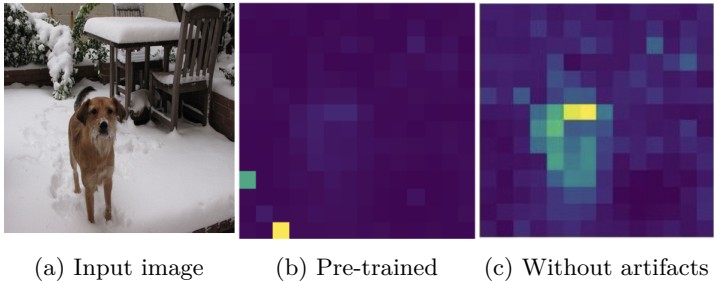

    (a) Input image      (b) Pre-trained      (c) Without artifacts

Figure 11: Attention map of DeiT-III at the last block, showing smoother attention after manual artifact removal.

## 5 Discussion

**Support for the original claims**
The experimental results of our study support most of the claims presented in the original paper, confirming the effectiveness of register tokens in removing high-norm artifact tokens and improving the interpretability of attention maps in ViT. Specifically, our findings align with the claims that high-norm tokens emerge in redundant, low-informative regions of the input image and that they primarily store global information. We observed that the introduction of register tokens successfully eliminated these artifacts without degrading classification and segmentation performance.

**Artifacts observed in smaller models**
Our study identified one inconsistency with the original claims. Contrary to the claim that artifacts emerge only in sufficiently large ViT models, we observed high-norm tokens in the DeiT-III Small model, which has significantly fewer parameters compared to the larger models evaluated in the original work. This finding challenges the assumption that artifact formation is strictly tied to model size and suggests that other factors, such as architectural design, may also contribute to their emergence. Although this observation highlights a potential gap in the original study's scope, it shows the importance of evaluating artifact behavior across a broader range of model configurations.

**Fine-tuning and limitations**
Due to computational constraints, we focused on fine-tuning pre-trained models rather than training models from scratch, as was done in the original study. This approach allowed us to investigate the impact of registers given different settings, such as the number of layers frozen or the regularization hyperparameter. Specifically, we introduced a regularization term, applying an L2-norm penalty during fine-tuning in order to remove high norm tokens. While our results confirmed a partial elimination of artifacts and an improvement in interpretability for fine-tuned models (in first epochs), the attention maps in `[CLS]` were not consistently smooth as training progressed (beyond 10 epochs). This suggests that a further exploration of the penalization term and the number of fixed layers is necessary.
Furthermore, large-scale experiments on diverse datasets and downstream tasks would be required to fully validate the scalability and robustness of the proposed method. Training a model from scratch to analyze artifact evolution throughout training could provide deeper insights; in particular, training a smaller version

of DeiT-III could help determine whether artifact tokens emerge even in a more compact variant of the architecture. However, given computational limitations, we leave this for future work.

**Conclusion**

The clarity of the original paper facilitated our reproduction of experiments. However, the lack of open-source code posed a challenge, requiring us to re-implement portions of the study from scratch. Additionally, the high computational cost of training large models influenced our decision to focus on fine-tuning rather than full-scale training.

Overall, our findings confirm the claims from the original paper, except for the presence of artifacts in DeiT-III Small, which suggest the need of further investigation into the conditions that lead to artifact emergence. To address this, we proposed a fine-tuning pipeline as a potential solution for artifact removal, resulting in smoother and more interpretable attention maps. Despite some limitations, our study demonstrates the effectiveness of register tokens in enhancing ViT interpretability and highlights fine-tuning as a resource-efficient strategy for mitigating these challenges.

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

# 6 Appendix

## 6.1 Fine-tuning: register tokens attention values

To investigate whether global information is being transferred to the register tokens, attention maps are visualized in Figure 14. An interesting observation is that all register tokens, despite having different attention patterns in the initial blocks, converge to a similar attention distribution in the final block. We hypothesize that this convergence occurs because the model is pre-trained, making it more challenging for register tokens to maintain diversity. In this context, we suggest that a single register token may suffice to reduce artifacts and smooth out the attention maps which aligns with the authors' findings.

## 6.2 Manual removal of artifacts

In this section we are going to manually replace artifacts from intermediate layers, corresponding to tokens with a high norm and high attention value. These artifacts are replaced with the minimum value from the immediate neighborhood.

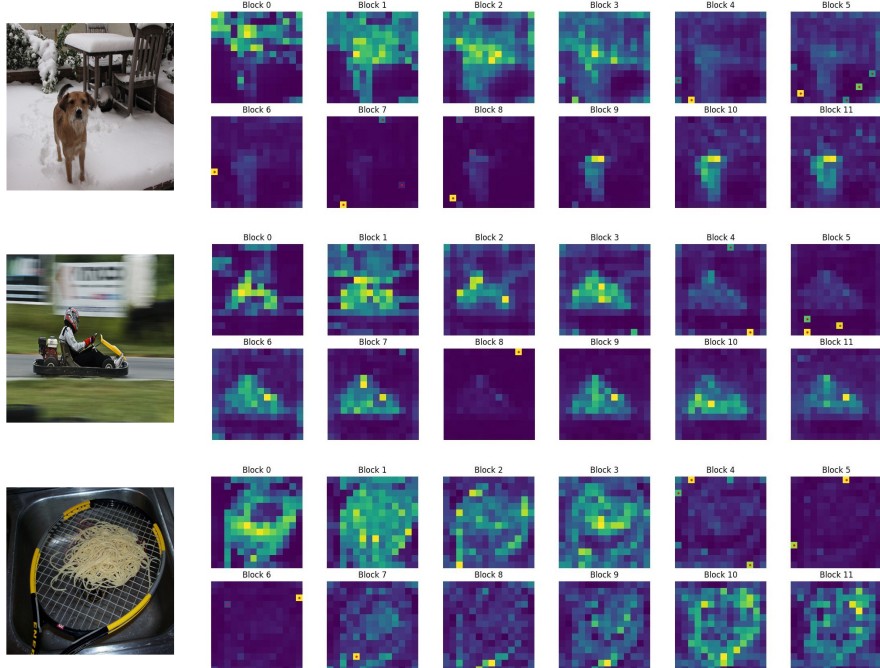

Figure 12: Attention progression for different images in DeiT-III small. The red dots represent removed tokens, identified as the intersection between top norm tokens and top attention tokens.

The resulting attention maps visible in Figure 12 are smoother and more interpretable, and the artifacts have disappeared. However, we observe a critical fall in the performance of the [CLS] token on classification tasks. This is in-line with the observation made of Vision Language Models in (2; 15) , showing that these artifacts do carry rich global information that is essential for model performance.
We thus showed that we can obtain cleaner attention maps without any kind of fine tuning or register addition. The investigation into the applicability of this method is out of the scope of this paper and is left for future work.

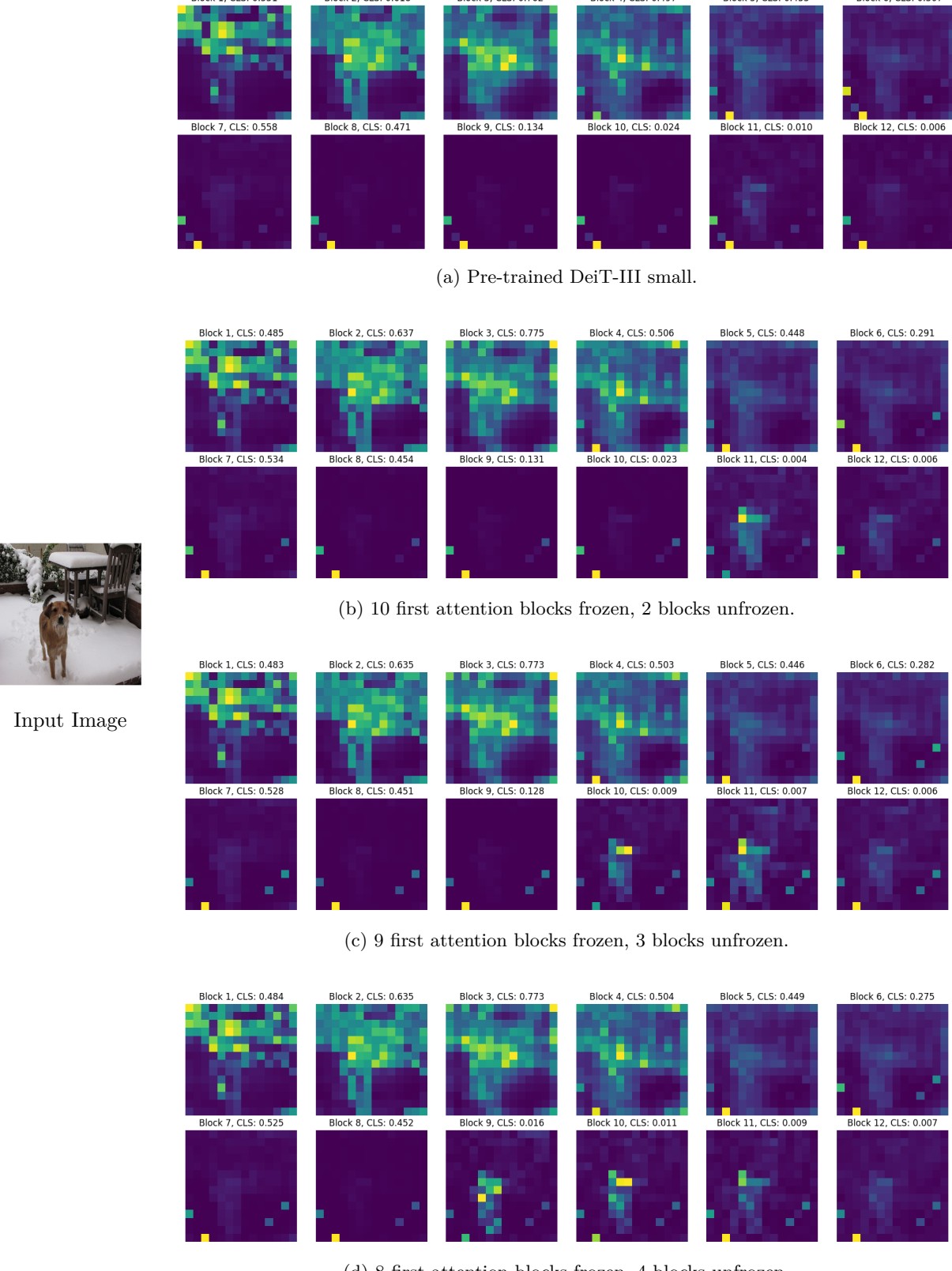

(a) Pre-trained DeiT-III small.

(b) 10 first attention blocks frozen, 2 blocks unfrozen.

Input Image

(c) 9 first attention blocks frozen, 3 blocks unfrozen.

(d) 8 first attention blocks frozen, 4 blocks unfrozen.

Figure 13: Fine-tuned DeiT-III small model with registers and varying number of frozen attention blocks. On top of every attention block visualization, the feature map norm of the [CLS] token has been computed.

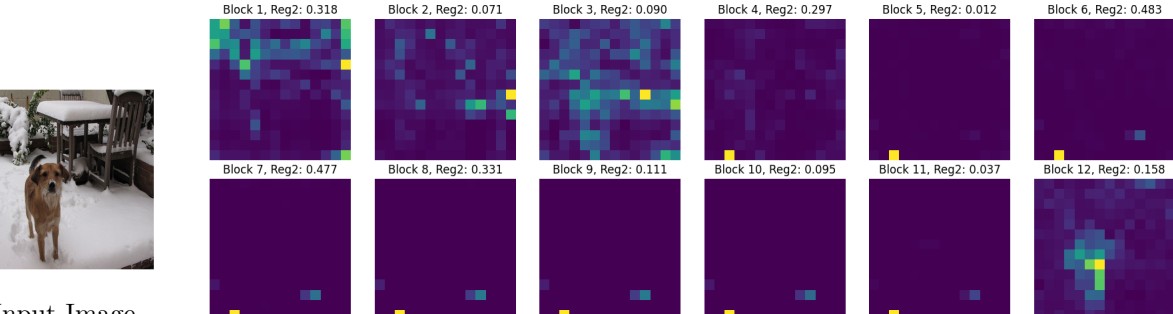

Figure 14: DeiT-III S: attention progression of an example register token (9 layers frozen, 4 registers, $\lambda = 0.1$).

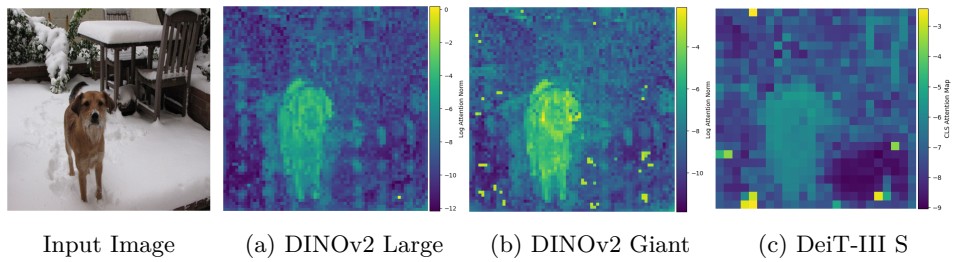

Input Image          (a) DINOv2 Large          (b) DINOv2 Giant          (c) DeiT-III S

Figure 15: Log-scale attention maps for DINOv2 Large, Giant, and DeiT models, allowing for better visualization of attention distribution.

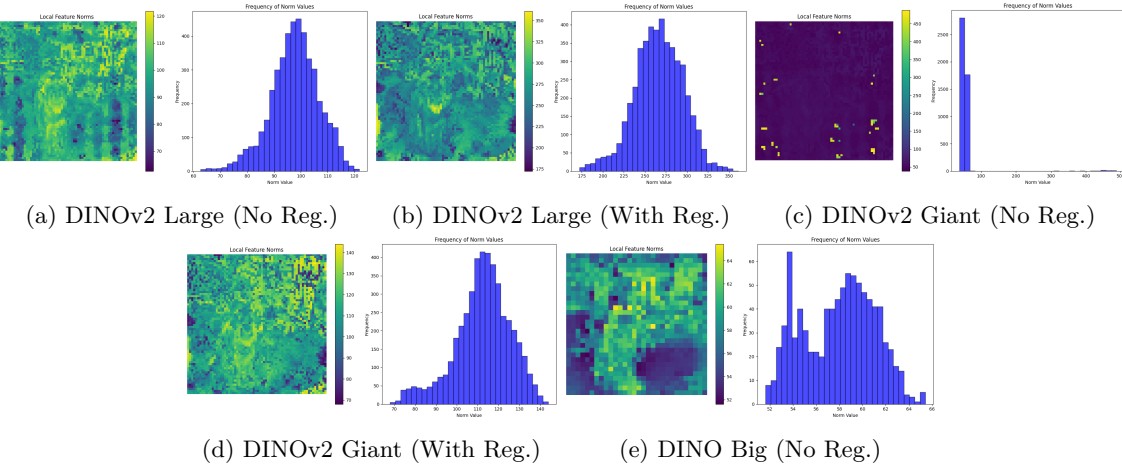

(a) DINOv2 Large (No Reg.)          (b) DINOv2 Large (With Reg.)          (c) DINOv2 Giant (No Reg.)

(d) DINOv2 Giant (With Reg.)          (e) DINO Big (No Reg.)

Figure 16: Feature norm distributions for DINO Big, DINOv2 Large, and DINOv2 Giant models, with and without register tokens. High-norm outliers are evident in models without registers, while their presence is reduced when register tokens are included.

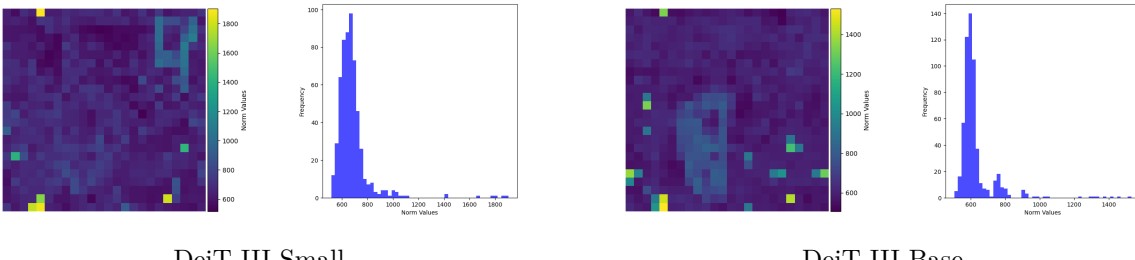

DeiT-III Small                              DeiT-III Base

Figure 17: Feature norms comparison between DeiT-III Small and Base models. High-norm tokens are observed near the edges of the image.

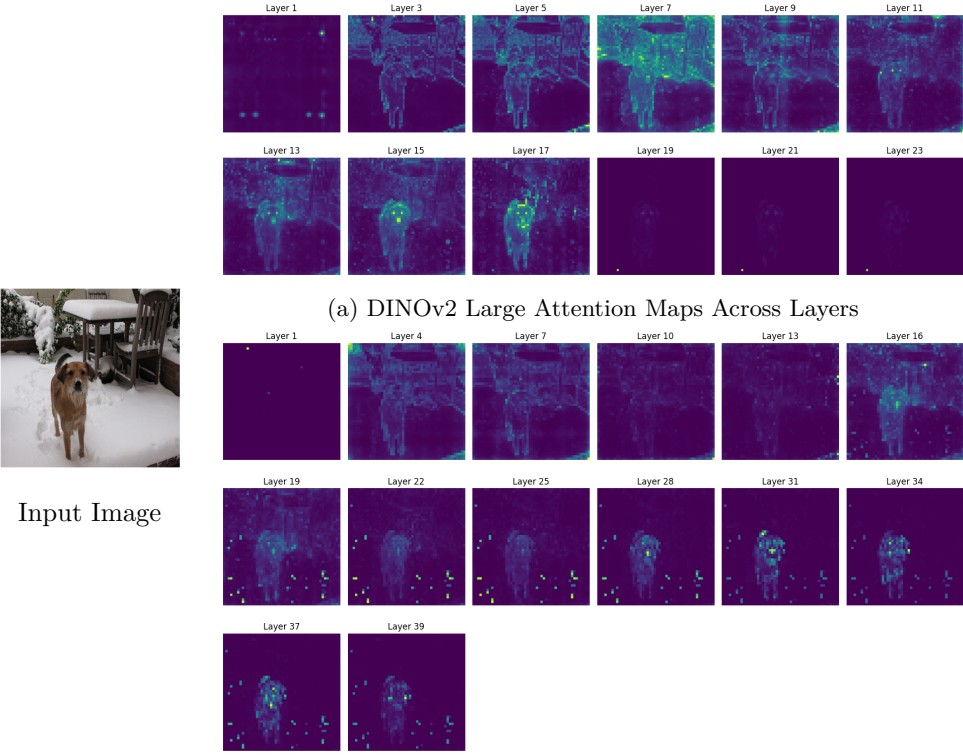

(a) DINOv2 Large Attention Maps Across Layers

Input Image

(b) DINOv2 Giant Attention Maps Across Layers

Figure 18: Attention maps across layers for DINOv2 Large and Giant models. For DINOv2 Giant, artifacts emerge around layers 15–16 out of 40 and exhibit global attention in earlier layers, covering the entire image before becoming more localized.

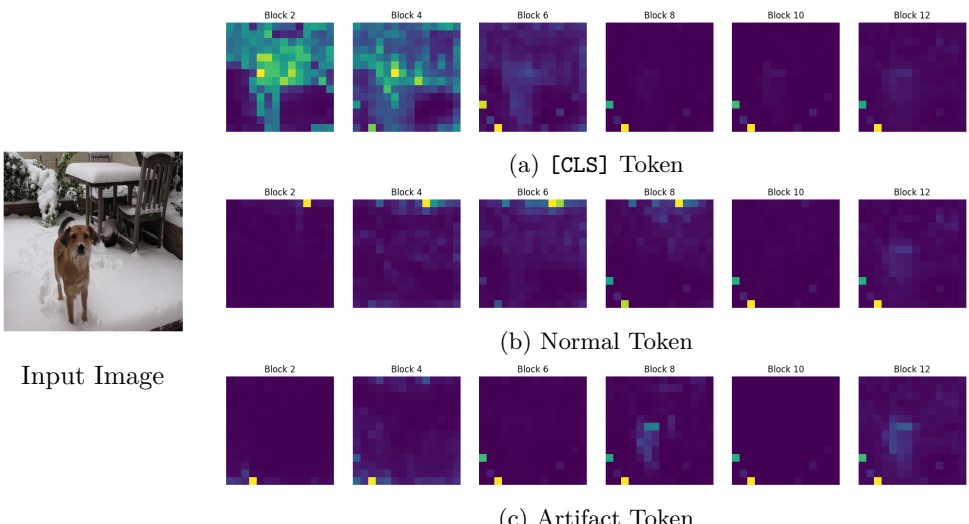

Figure 19: Attention maps for DeiT-III S, showing the `[CLS]` token, a randomly selected normal patch token, and an artifact token across layers. Artifacts appear around block 5 out of 12. The `[CLS]` and artifact tokens exhibit attention on the main subject, whereas normal patch tokens do not.

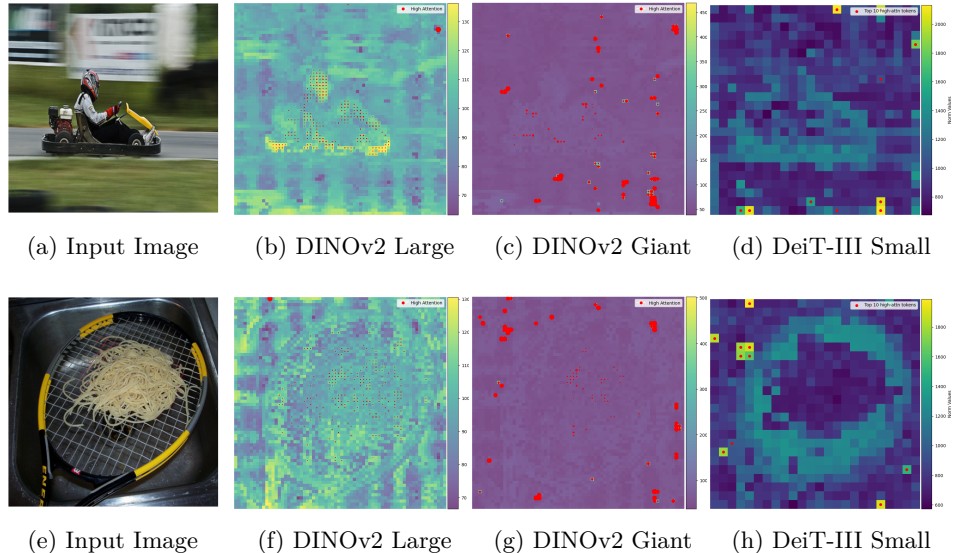

Figure 20: High-attention patches and feature norms for DINOv2 and DeiT-III models with different input images. High-norm tokens align with high-attention tokens, confirming their artifact nature. Red dot size in the second and third plots indicates attention magnitude.

**Without Registers**

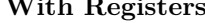

(a) Input Image    (b) DINO B    (c) DINOv2 L    (d) DINOv2 G    (e) DeiT-III S    (f) DeiT-III B

**With Registers**

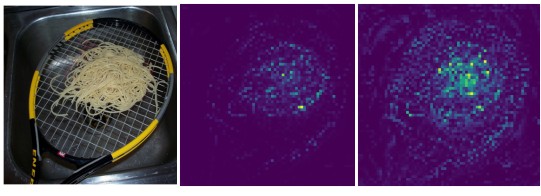

(g) Input Image    (h) DINOv2 L    (i) DINOv2 G

**Without Registers**

(j) Input Image    (k) DINO B    (l) DINOv2 L    (m) DINOv2 G    (n) DeiT-III S    (o) DeiT-III B

**With Registers**

(p) Input Image    (q) DINOv2 L    (r) DINOv2 G

Figure 21: Attention maps for DINO, DeiT-III Small/Base, and DINOv2 Large/Giant with and without register tokens on an additional ImageNet image. Results confirm that artifacts appear in all models except DINO Base and are mitigated by register tokens.

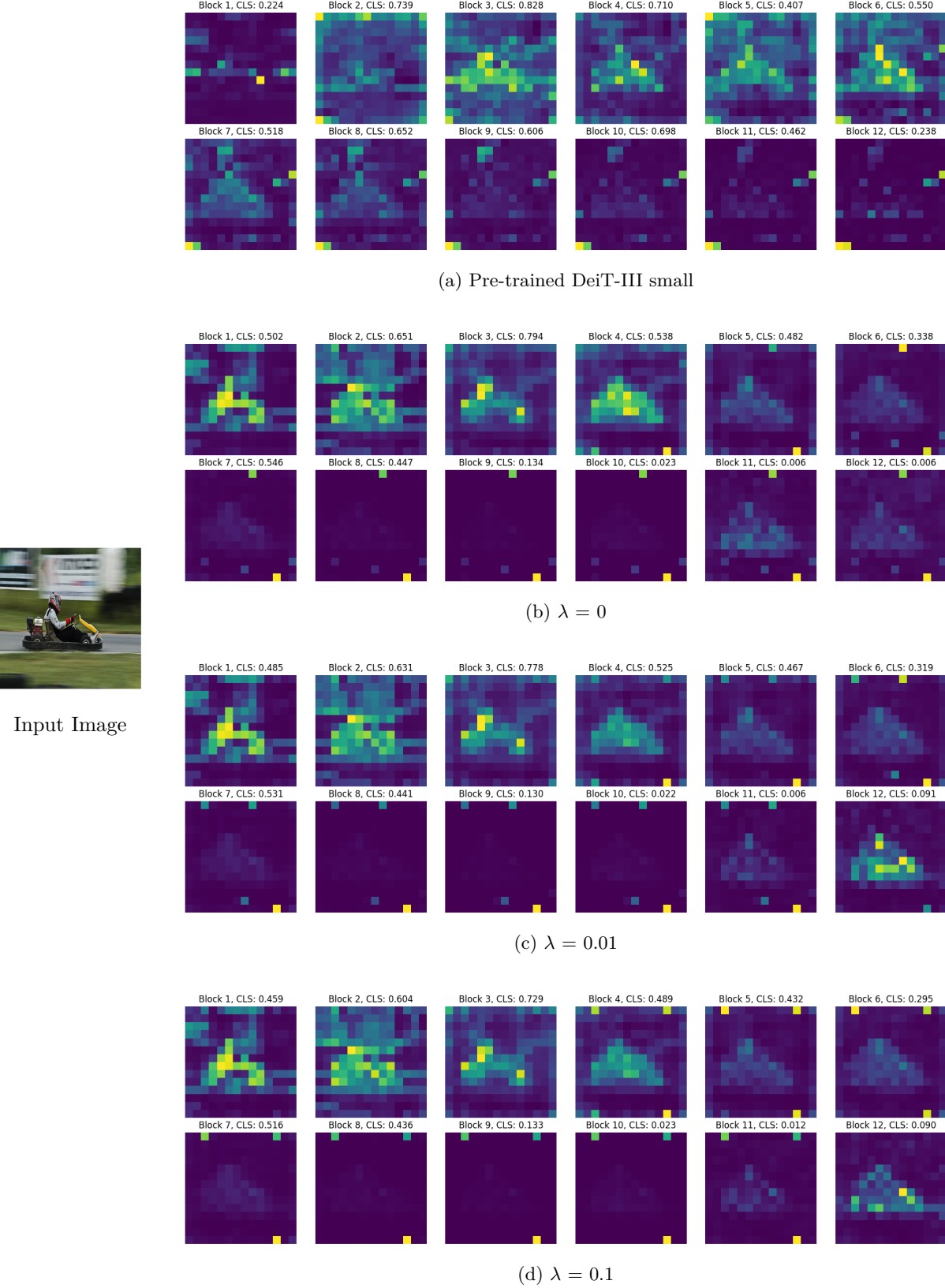

Figure 22: Fine-tuned DeiT-III small model with 4 registers, first 10 attention blocks frozen and 2 blocks unfrozen, and different L2-norm $\lambda$ parameters. The higher the $\lambda$ parameter, the more high-norm tokens are penalized. On top of every attention block visualization, the feature map norm of the [CLS] token has been computed.

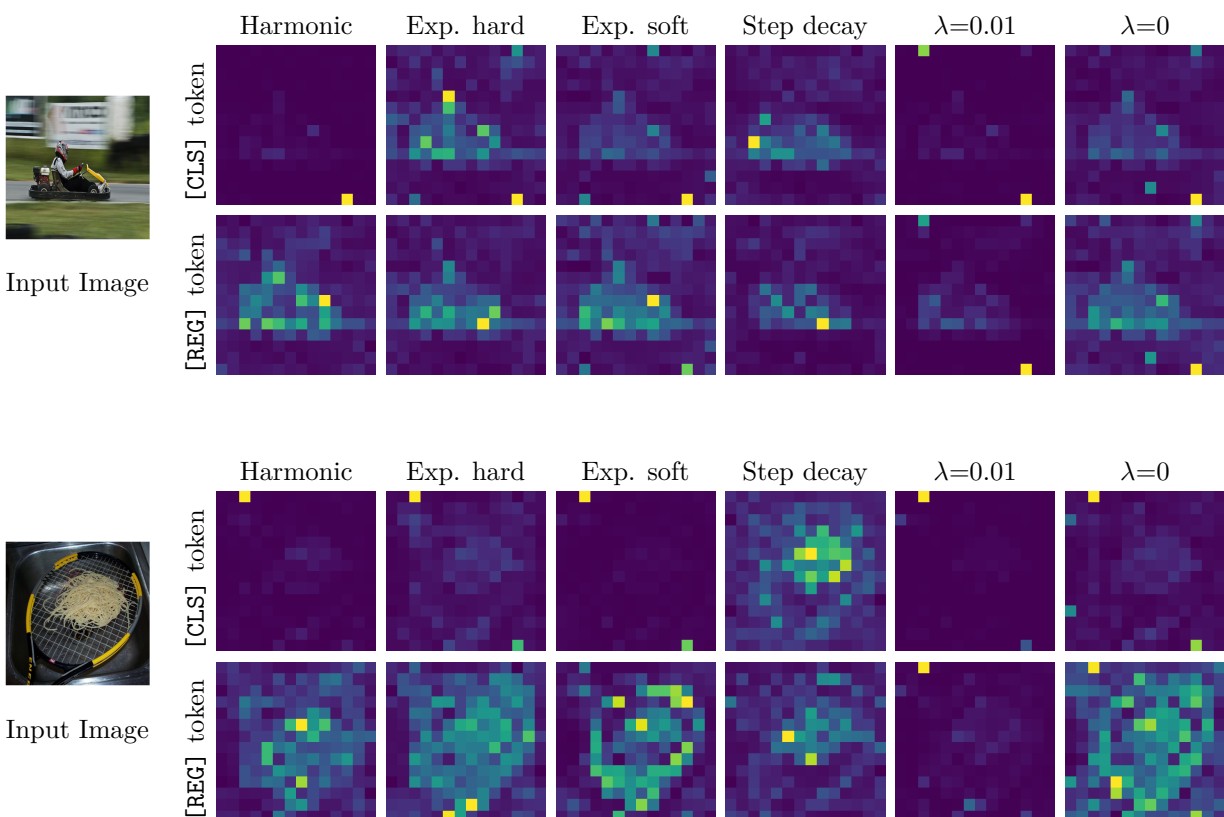

Figure 23: Comparison of `[CLS]` and the register (`[REG]`) tokens' attention values from the last attention block of a fine-tuned DeiT-III small model after 50 epochs using different lambda schedules: harmonic decay ($\lambda(0) = 0.1$), hard exponential decay ($\lambda(0) = 0.1$), soft exponential decay ($\lambda(0) = 0.01$), step decay ($\lambda(0) = 0.01$), $\lambda = 0.01$, and $\lambda = 0$.

