# OpenReview forum: "Reproducibility Study of “Vision Transformers Need Registers”"
_TMLR — Rejected by TMLR_

### Review · Reviewer_Y5Tz · 2025-03-31

**Summary Of Contributions:**

This paper reproduces and validates the findings of "Vision Transformers Need Registers", which addresses the issue by incorporating register tokens—learnable placeholders added to the input sequence—that mitigate artifacts and produce smoother feature maps. Authors evaluate the presence of artifacts in DeiT-III and DINOv2 architectures. The findings align with the claims that high-norm tokens emerge in
redundant, low-informative regions of the input image and that they primarily store global information.

**Audience:**

Yes

**Broader Impact Concerns:**

no significant ethical concerns are identified in the work.

**Claims And Evidence:**

Yes

**Requested Changes:**

This paper reproduces part of the original work and demonstrates the effectiveness of incorporating register tokens. However there are some missing components which may diminish its overall contribution.

1. As shown in the original work, other models with different pretraining paradigms (e.g., OpenCLIP) could be incorporated into the experiments to systematically examine the proposed claims.

2. Segmentation and depth estimations tasks could be added to evaluate how register tokens work for different tasks.

3. An investigation into how the number of register tokens affects the performance of various downstream tasks could be incorporated.

4. In Table 5, a list of various hyper-parameters is provided, but it is unclear what these numbers are intended to demonstrate.

**Strengths And Weaknesses:**

**Strengths**
1. Authors reproduce most of the claims proposed in the original paper.
2. Extensive experiments and visualizations demonstrate the effectiveness of incorporating register tokens.

**Weaknesses**

Overall, this paper primarily focuses on reproducing previously published work, essentially serving as a simple verification study. It appears to offer only limited additional baselines, analysis, ablations, or insights.
1. In Figure 2, why are there five models without register tokens but only two models with register tokens?
2. This reproductions mainly focuses on the classification tasks, while the original paper conducts experiments on both segmentation and classification tasks.
3. The use of DINO v2 and DeiT-III relies on CLS tokens, which require supervised fine-tuning on a labeled dataset. This choice limits the reproduction’s scope, as other model types with different pretraining paradigms —such as CLIP or MAE—might provide stronger evidence.
4. In Table 5, a list of various hyper-parameters is provided, but it is unclear what these numbers are intended to demonstrate.

---

> ### Author Response · Authors · 2025-04-15
> **Addressing the requested changes and questions in the review**
>
> We thank the reviewer for their remarks and thoughtful review. We answer the questions below:
>
> `In Figure 2, why are there five models without register tokens but only two models with register tokens?`
>
> The reason why Figure 2 includes five models without register tokens but only two with register tokens is that pre-trained versions of DeiT-III models with register tokens are not publicly available. For fairness and consistency, we limited the comparison in Figure 2 to publicly released checkpoints. However, we do include attention maps for fine-tuned DeiT-III models with register tokens later in the paper (see Figure 13), which we fine-tuned ourselves. We have clarified this point in the caption of Figure 2 to avoid confusion.
>
> `As shown in the original work, other models with different pre-training paradigms (e.g., OpenCLIP) could be incorporated into the experiments to systematically examine the proposed claims.`
>
> `Segmentation and depth estimations tasks could be added to evaluate how register tokens work for different tasks.`
>
> Our work primarily focuses on DINO and DeiT models, as they provide strong baselines for evaluating the role of register tokens. Due to computational constraints, we were unable to replicate our full set of experiments on additional models such as OpenCLIP. However, we agree that extending our analysis to models with contrastive language-image pre-training would be valuable. We see this as a direction for future work to conduct a more in-depth study on OpenCLIP.
> Considering the second part, we included experiments in Section 4.1.8 to evaluate the effect of register tokens on a dense prediction task. Specifically, we measured mIoU on the ADE20K dataset using both DINOv2-L and DeiT-III-Small backbones, with and without register tokens. We chose to focus on semantic segmentation as a representative dense prediction task due to computational constraints.
>
> `An investigation into how the number of register tokens affects the performance of various downstream tasks could be incorporated.`
>
> For DINOv2, we used the publicly available backbone pretrained with 4 register tokens, and therefore did not retrain models from scratch with different numbers of registers. As a result, we report results for 0 and 4 register tokens in all downstream experiments involving DINOv2.
> For DeiT-III Small, we conducted fine-tuning experiments with 0, 1, and 4 register tokens and with 0, 1, 2, 4 for the segmentation task. All results are provided in the paper.
>
> `In Table 5, a list of various hyper-parameters is provided, but it is unclear what these numbers are intended to demonstrate.`
>
> We have updated Table 5 (now Table 8) to show the comparison between the performance of adding 4 registers, to stay consistent with the original paper and included additional comments on the results. The addition of registers preserves the model’s core behavior and shows no deterioration in classification performance.

---

### Review · Reviewer_rKBu · 2025-04-06

**Summary Of Contributions:**

This paper presents a reproducibility study of the paper "Vision Transformers Need Registers", empirically verifying several key claims made in the original study. Specifically, it validates: 1) the presence of high-norm artifact tokens in the DeiT-III and DINO-v2 models, 2) the similarity between these artifact tokens and their neighboring tokens, as well as their encoding of global image information and lacking of local details 3) the ability of register tokens to mitigate these high-norm artifact tokens. In contrast to the original findings, this study observes artifact tokens even in smaller ViT models, such as DeiT-III Small. Additionally, this research explores an alternative approach to addressing artifact tokens through fine-tuning pre-trained ViTs, rather than the original method of re-training a ViT with register tokens from scratch.

**Audience:**

Yes

**Broader Impact Concerns:**

Currently, I do not identify any ethical concerns in the paper.

**Claims And Evidence:**

Yes

**Requested Changes:**

Please refer to the aforementioned weaknesses, particularly the first two, as they are crucial for evaluating the contribution of this work.

**Strengths And Weaknesses:**

**Strengths**
- This work comprehensively investigates the key claims of "Vision Transformers Need Registers" with extensive experiments.
- This work demonstrates that fine-tuning with added register tokens can effectively eliminate artifact tokens, which is more efficient than the original method which re-trains the entire model from scratch. Based on the proposed fine-tuning framework, the authors also investigate a variety of configurations, including: 1) the incorporation of L2-normalization, 2) the number of register tokens and 3) the layers to fine-tune.

**Weaknesses**
- The paper observes that artifact tokens also emerge in smaller models such as DeiT-III Small, which contradicts the claim of the original paper. However, it fails to provide a solid explanation for this phenomenon.
- Although the attention map visualization suggests that the fine-tuning method produces smoother and more interpretable attention maps, there is no quantitative evaluation or comparison with the original training-from-scratch approach, in terms of computational cost and the mitigation of artifact tokens.
- The description of computational requirements in Section 3.5 lacks clarity. For instance, the authors state that *"For the DINO and DINOv2 experiments, computations were performed on a CPU for approximately 15 hours."* However, it is unclear which specific experiments this refers to, and the sizes of the DINO and DINOv2 models used are not specified. To improve clarity, it would be helpful to include a table summarizing the computational cost associated with each individual experiment.
- Minor Issues:
  - The font size used in the figures is relatively small; increasing it would improve readability.
  - Including the names of the benchmarks in the table captions would provide greater clarity and context.

---

> ### Author Response · Authors · 2025-04-15
> **Addressing the requested changes and questions in the review**
>
> We thank the reviewer  for their remarks and thoughtful review. We answer the questions below:
>
> `The paper observes that artifact tokens also emerge in smaller models such as DeiT-III Small, which contradicts the claim of the original paper. However, it fails to provide a solid explanation for this phenomenon.`
>
> In our study, we observe that artifact tokens also emerge in smaller models, such as DeiT-III Small. This directly contradicts the original paper's claim, based only on DINOv2, that outliers appear only in the largest models (Large, Huge, and Giant). The original paper suggests that both the pretraining paradigm and model size influence the emergence of artifacts. However, our findings show that artifacts are not limited to large models. This indicates that their appearance is not strictly dependent on model size. While a complete explanation remains open, our work advances the understanding of the phenomenon by showing that it generalizes across model scales. Further analysis of the internal mechanisms is left for future work, as this paper primarily focuses on reproducibility and the proposal of efficient alternative solutions.
>
> `Although the attention map visualization suggests that the fine-tuning method produces smoother and more interpretable attention maps, there is no quantitative evaluation or comparison with the original training-from-scratch approach, in terms of computational cost and the mitigation of artifact tokens.`
>
> To address this gap in our analysis, we provide a quantitative comparison of token norms before and after fine-tuning. Specifically, we include histograms (Figure 9) showing how the distribution of token L2 norms changes: fine-tuning removes the bimodal behavior observed in the training-from-scratch setting, reducing the presence of artifact tokens (defined as outlier that typically appear in uninformative background areas).
>
> `The description of computational requirements in Section 3.5 lacks clarity. For instance, the authors state that "For the DINO and DINOv2 experiments, computations were performed on a CPU for approximately 15 hours." However, it is unclear which specific experiments this refers to, and the sizes of the DINO and DINOv2 models used are not specified. To improve clarity, it would be helpful to include a table summarizing the computational cost associated with each individual experiment.`
>
> To improve clarity, we have revised Section 3.5 to include a more detailed breakdown of the computational requirements for both the DINO and DeiT-III experiments. We now explicitly specify which experiments were run on CPU or GPU, and clarify the model sizes involved. Additionally, we have added a summary table that outlines the compute time and hardware configuration for each major experimental setup, including the fine-tuning runs. This aims to make the computational costs more transparent and easier to interpret.
>
> `Minor Issues:
> The font size used in the figures is relatively small; increasing it would improve readability.
> Including the names of the benchmarks in the table captions would provide greater clarity and context.`
>
> The table captions have been updated to explicitly include the names of the benchmarks used, providing greater clarity and context.

---

### Review · Reviewer_NpsV · 2025-04-09

**Summary Of Contributions:**

The paper presents a reproducibility study of "Vision Transformers Need Registers," which proposes adding register tokens to the image token sequence during vision transformer pre-training. These register tokens, independent of the input image, are designed to address the emergence of high-norm artifact tokens during pre-training.

The empirical findings in the reproducibility study largely align with the original paper's conclusions. However, two differences were identified: First, the authors found that fine-tuning with register tokens under L2 normalization could partially mitigate the artifact token problem, offering an alternative to pre-training from scratch with register tokens. Second, they discovered that artifact tokens can emerge in smaller models, contrary to the original paper's findings.

**Audience:**

Yes

**Claims And Evidence:**

Yes

**Requested Changes:**

Please refer to Weaknesses above.

**Strengths And Weaknesses:**

## Strengths
- The writing is easy to follow
- The paper presents empirical results across different model families and scales to support their arguments
- Code for this reproducibility study is/will be publicly released

## Weaknesses
------------------------------------------------
In 4.1.3 High-norm high-attention correspondence:
>For DINOv2 models, high-attention outlier tokens were defined as the top 5% of tokens with the highest
attention values [...]  Similarly, for the DeiT-III model, high-attention tokens, selected as the top 10 tokens with the highest
attention values [...]

and in 4.1.4 Cosine similarity between high-norm tokens and their neighbors:
>  Cosine similarity analysis on DINOv2 Large revealed that high-norm tokens, defined as the top 15% output tokens with high norms [...]

Why do the definitions for 'outlier tokens' differ between models and sections?

------------------------------------------------
It would be helpful if the authors could provide more details about some experimental settings. For example,
in 4.1.6 High-norm tokens hold little local information:
> To evaluate the local information stored in high-norm tokens, we trained a linear classifier to predict the position
of each patch token in the image.

How was this done exactly?

------------------------------------------------

Although experiments are conducted for different model families, more discussion specifically on why different models lead to different results would be appreciated. For example:

- Fig 2: DINOv2 G appears to be less affected by artifact tokens compared to DINOv2 L, despite having a larger model size, while DINOv2 models are less affected in comparison to the DeiT-III models.
- The study suggests that artifact tokens can emerge in smaller models too -- a different finding from the original paper, why is this the case?

---

> ### Author Response · Authors · 2025-04-15
> **Addressing the requested changes and questions in the review**
>
> We thank the reviewer for their remarks and thoughtful review. We answer the questions below:
>
> `In 4.1.3 High-norm high-attention correspondence: For DINOv2 models, high-attention outlier tokens were defined as the top 5% of tokens with the highest attention values [...] Similarly, for the DeiT-III model, high-attention tokens, selected as the top 10 tokens with the highest attention values [...]
> and in 4.1.4 Cosine similarity between high-norm tokens and their neighbors:
> Cosine similarity analysis on DINOv2 Large revealed that high-norm tokens, defined as the top 15% output tokens with high norms [...]
> Why do the definitions for 'outlier tokens' differ between models and sections? `
>
> We've revised the paper to clarify these sections. In section 4.1.3, we do a qualitative analysis to show the correspondence between high-norm and high-attention tokens. For this we use the top-5% of high-attention tokens.
> In both sections, we have used a unified notion and defined the “outlier tokens” as top-5% of tokens with the highest norm.
>
>
> `It would be helpful if the authors could provide more details about some experimental settings. For example, in 4.1.6 High-norm tokens hold little local information:
> To evaluate the local information stored in high-norm tokens, we trained a linear classifier to predict the position of each patch token in the image.
> How was this done exactly?`
>
> To evaluate whether high-norm tokens retain local information, we trained a linear classifier to predict the original spatial position of each patch token. We extracted token embeddings and their corresponding 2D positions from CIFAR-10 images, and split them into high-norm and normal-norm groups based on their L2-norm. A separate linear classifier was trained on each group to predict the position of each patch token in the image. We reported top-1 accuracy and average positional distance across three random seeds.
> This procedure allowed us to quantify how much spatial information is retained in the token embeddings. As suggested, we updated the paper to include a description of this setup in Section 4.1.6 for clarity.
>
> `Although experiments are conducted for different model families, more discussion specifically on why different models lead to different results would be appreciated. For example:
> Fig 2: DINOv2 G appears to be less affected by artifact tokens compared to DINOv2 L, despite having a larger model size, while DINOv2 models are less affected in comparison to the DeiT-III models.
> The study suggests that artifact tokens can emerge in smaller models too -- a different finding from the original paper, why is this the case?`
>
> In our study, we observe that artifact tokens also emerge in smaller models, such as DeiT-III Small. This directly contradicts the original paper's claim, based only on DINOv2, that outliers appear only in the largest models (Large, Huge, and Giant). The original paper suggests that both the pretraining paradigm and model size influence the emergence of artifacts. However, our findings show that artifacts are not limited to large models. This indicates that their appearance is not strictly dependent on model size. While a complete explanation remains open, our work advances the understanding of the phenomenon by showing that it generalizes across model scales. Further analysis of the internal mechanisms is left for future work, as this paper primarily focuses on reproducibility and the proposal of efficient alternative solutions.

---

### Decision · Action_Editor_Lwf4 · 2025-05-12

**Recommendation:** Reject

**Comment:**

This paper makes two main contributions: a reproducibility study that largely confirms the results of "Vision Transformers Need Registers", and two new empirical observations. Reviewers are generally supportive of these contributions and acknowledge the effectiveness of their fine-tuning method to mitigate the artifact token issue.

However, two reviewers noted that the claim regarding "artifact tokens emerging in smaller ViT models" is not sufficiently explained or verified, particularly with respect to its generalizability. Relevantly, an experiment on comparing fine-tuning and pretraining with register tokens, which was requested during the rebuttal but remains absent following the rebuttal.

This observation is significant, as it contradicts a key claim from the original paper. While the authors' finding may have merits, the reviewers indicated that the analysis lacks depth, and the generalizability of the result remains unclear. The AE agrees with the reviewers’ concern and believes that, although the paper shows potential, it is premature to accept the submission without addressing this concern through deeper analysis, more empirical validation, and tests of generalizability.

**Audience:**

The target audience of this paper is similar to that of Vision Transformers Need Registers who are researchers studying and improving Vision Transformers.

**Claims And Evidence:**

The paper verifies the five claims made in "Vision Transformers Need Registers" using three smaller-scale datasets: CIFAR, ImageNet1k, and ADE20K. The authors empirically find that most of the original claims are supported.

Additionally, the paper presents two new observations:

* Fine-tuning with L2-norm regularization on register tokens can partially mitigate the artifact token issue, which offers an alternative approach to that proposed in the original paper.

* Artifact tokens can also emerge in smaller models, which contrasts with the findings of the original study.

These observations are backed by results from their experimental setup.

**Resubmission Of Major Revision:**

The authors may consider submitting a major revision at a later time.